# CODE DIFFUSION MODELS ARE CONTINUOUS HUMAN NOISE OPERATORS

## ABSTRACT

Diffusion for code generates code by iteratively removing noise from the latent representation of a code snippet. During later steps of the diffusion process, when the code snippet has almost converged, differences between discrete representations of these snippets look like last-mile repairs applied to broken or incomplete code. We evaluate the extent to which this resemblance can be exploited to leverage pre-trained code diffusion models for the problem of last-mile repair by considering two applications with significant potential. First, we can leverage the diffusion model for last-mile repair by adding noise to a broken code snippet and resuming the diffusion process. Second, we can leverage the diffusion model to generate an arbitrary amount of training data for other last-mile repair approaches (that are computationally more efficient) by sampling an intermediate program (input) and the final program (output) from the diffusion process. We perform experiments on three domains (Python, Excel and PowerShell) to evaluate both applications, as well as analyze properties. [1]

## 1 INTRODUCTION

Diffusion models have emerged as a powerful paradigm in generative modeling, particularly for tasks that involve complex data structures (Ho et al., 2020). Instead of generating a sample from a distribution in one go (like a GAN or VAE) or auto-regressively (like a GPT) they learn to iteratively reverse diffusion steps that add (typically Gaussian) noise to the data. Initially popularized in the domain of image generation, diffusion models have since been adapted for modalities like video generation (Ho et al., 2022; Xing et al., 2023)—which requires a temporal component—and text or code generation (Li et al., 2022; Singh et al., 2023a)—which requires diffusion over discrete tokens.

One approach of applying diffusion to discrete domains, like text or code, involves embedding the input, performing diffusion in the embedded representation, and projecting the denoised embeddings back to discrete tokens. To train this model end-to-end, the loss incorporates a component over the discrete tokens, meaning that representation from each step of the reverse diffusion process can be converted back to the discrete space (Lin et al., 2023). During initial generations, decoding the latent representation does not resemble anything and tokens frequently change, but in later generations, these decoded representations become readable and it takes multiple steps to change one token.

As an example, consider the following generations from pre-trained CodeFusion (Singh et al., 2023a) models—without natural language conditioning—trained on Excel

```
     (t_75%)    =IF(COUNTIF(A:A, '>10')=0, 'No values', AVERAGE(A:A))
(t_90% - t_100%)  =IF(COUNTIF(A:A, '>10')=0, 'No values', AVERAGEIF(A:A, '>10'))
```

and Python

```
(t_75% - t_90%)   words = read('myfile').split()
      (t_100%)    words = open('myfile').read().split()
```

with changed tokens highlighted in red. It appears as if the diffusion model can look at the whole (discrete) program, determine what is missing to make it functional, and apply those fixes. This is

---

[1] The code and associated datasets can be found at `redacted`

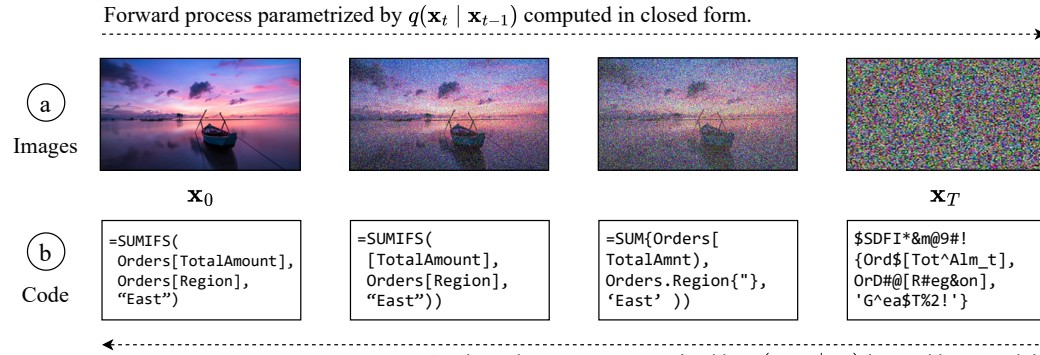

Figure 1: Example of diffusion for (a) images and (b) code. Pure $\mathbf{x}_T$ is iteratively denoised into a sample $\mathbf{x}_0$ from the target distribution by a model trained on data from the forward process.

exactly the premise of last-mile repair, in which the goal is to repair broken code in such a way that the solution differs minimally from the broken code (Bavishi et al., 2022). A major challenge in training (last-mile) repair systems is the long-tail problem in obtaining training data (Huang et al., 2023) and out-of-distribution generalization when introducing synthetic errors (Joshi et al., 2024).

In this paper, we address those challenging by asking if **code diffusion can serve as a continuous human noise operator?** In other words, we evaluate the extent to which the discrete representations obtained during reverse diffusion steps—which *look* like mistakes that humans could make—are representative of mistakes that humans actually make. This exploration has two main applications: we can use the diffusion model to **directly repair code**, and we can use the diffusion model to **generate training data** for specialized approaches.

We support our claims with experiments on three programming languages: Python, PowerShell and Excel. We find that diffusion models are capable of last-mile repair, with the models being able to repair 56.4–68.2% of Python and Excel snippets across different noise levels. We also find that the diffusion-generated synthetic data has higher diversity and complexity compared to existing data generators and GPT-4o, which is reflected in higher performance observed (+2.5 – 3.5%) when fine-tuning different models (`codet5-small`, `phi-35-mini` and `mistral-7b`) on the synthetic data.

## 2 BACKGROUND

### 2.1 DIFFUSION MODELS

A diffusion model is a latent variable model that constructs a Markov chain $\mathbf{x}_0, \mathbf{x}_1 \cdots \mathbf{x}_T$ and simulates data $\mathbf{x}_0 \sim p_{\text{data}}$ by learning to reverse this Markov chain (Ho et al., 2020). The sequence of continuous latent variables $\mathbf{x}_{1:T}$ is constructed by incrementally adding (typically Gaussian) noise to data $\mathbf{x}_0$ until, at diffusion step $T$, samples $\mathbf{x}_T$ are approximately Gaussian. Each transition $\mathbf{x}_{t-1} \rightarrow \mathbf{x}_t$ is parametrized by $q(\mathbf{x}_t \mid \mathbf{x}_{t-1}) = \mathcal{N}(\mathbf{x}_t; \sqrt{1 - \beta_t}\mathbf{x}_{t-1}, \beta_t\mathbf{I})$ where the hyperparameter $\beta_t$ is the amount of noise added at diffusion step $t$. The diffusion model generates samples by reversing this chain: it iteratively denoises the sequence of latent variables $\mathbf{x}_{T:0}$ to approximate a sample from the target distribution. Each denoising transition $\mathbf{x}_t \rightarrow \mathbf{x}_{t-1}$ is parametrized by the model that predicts $p_\theta(\mathbf{x}_{t-1} \mid \mathbf{x}_t) = \mathcal{N}(\mathbf{x}_{t-1}; \mu_\theta(\mathbf{x}_t, t), \Sigma_\theta(\mathbf{x}_t, t))$. In practice, instead of constructing the whole chain, we can immediately obtain $\mathbf{x}_t$ from $\mathbf{x}_0$ as $\mathbf{x}_t = \sqrt{\bar{\alpha}_t}\mathbf{x}_0 + \sqrt{1 - \bar{\alpha}_t}\epsilon$ with $\bar{\alpha}_t = \prod_{i=1}^t 1 - \beta_t$ and $\epsilon \sim \mathcal{N}(0, \mathbf{I})$. The model $f_\theta(\mathbf{x}_t, t)$ is parametrized to predict $\mathbf{x}_0$ with an empirically validated loss function $\mathcal{L}_{\text{simple}} = \mathbb{E}_{\mathbf{x}_0, \epsilon_t, t}\|f_\theta(\mathbf{x}_t, t) - \mathbf{x}_0\|^2$ (Ho et al., 2020; Li et al., 2022). At inference time, we compute $\mathbf{x}_{t-1} = \sqrt{\bar{\alpha}_t}f_\theta(\mathbf{x}_t, t) + \sqrt{1 - \bar{\alpha}_t}\epsilon$ to iteratively denoise $\mathbf{x}_t$.

**Example 1** *Figure 1 shows the generations of a latent diffusion model. It can be seen how the model iteratively denoises to the concrete representation from the output space.*

## 2.2 DIFFUSION MODELS FOR CODE

Code generation is a discrete generation task, where the expected output is a snippet $\mathbf{c} = [c_1, \ldots, c_k]$ of $k$ tokens. CodeFusion (Singh et al., 2023a) draws inspiration from text diffusion (Li et al., 2022) where each token $c_i$ is embedded $\text{E}(c_i) \in \mathbb{R}^d$ to convert $\mathbf{c}$ into a continuous representation $\text{E}(\mathbf{c}) \in \mathbb{R}^{kd}$ to which a regular diffusion process can be applied. In the reverse process, a trainable rounding step $p_\theta(c_i \mid x_{\leq i})$ computes a distribution over possible tokens for each position $i$ given all previous (denoise) tokens $x_{<i}$. Note that the decoder is trained to always generate a constant number of $n > k$ tokens, one of which is an end-of-sequence token and $n - k - 1$ padding tokens. Like CodeFusion, we set $n = 128$.

**Example 2** *Figure 1 shows the generations of a latent code diffusion model. The intermediate representations, when visualized in the discrete token space, show how the model iteratively denoises to a syntactically valid Excel formula. Furthermore, we can see how the generation at $t_{75\%}$ has the table name missing in the structured reference which the model fixes through refinement.*

$$
\begin{array}{ll}
(t_{75\%}) & \texttt{=SUMIFS([TotalAmount], Orders[Region], "East"))} \\
(t_{90\%} - t_{100\%}) & \texttt{=SUMIFS(Orders[TotalAmount], Orders[Region], "East"))}
\end{array}
$$

More generally, Figure 2 shows trends in discrete code refinement over diffusion time-steps as (a) the number of tokens being changed and (b) the maximum distance between tokens being edited. In Figure 2a, as expected, significantly fewer tokens are changed further down the diffusion process. In Figure 2b, one key observation is that diffusion models tend to prioritize global repairs before drilling down and addressing local issues. These trends of fewer and localized edits near the end of the diffusion process motivate the application of the diffusion process for last-mile repair.

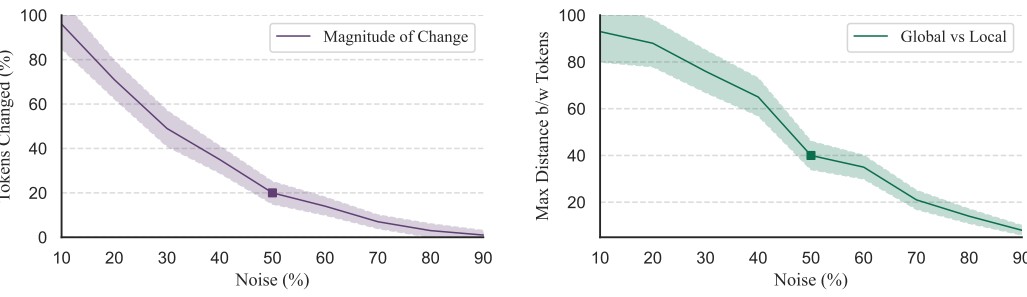

(a) Percentage of tokens changed in each iteration.       (b) Maximal distance between two edits.

Figure 2: Trends in code refinement over diffusion time steps.

## 3 DIFFUSION FOR REPAIR

Let $\hat{c}$ be a buggy code snippet that is not accepted by the compiler. The goal of last-mile repair is to find a code snippet $c^* = \arg\min_c d(c, \hat{c})$ such that $c^*$ is accepted by the compiler and performs a task intended by the user, with $d$ the edit distance between two code snippets. Like previous work on last-mile repair, we only consider syntactic errors (Bavishi et al., 2022; Joshi et al., 2023). In the following three sections, we respectively reiterate the components and training process of CODEFUSION, describe how to apply it to problem of last-mile repair, and describe how to generate pairs $(\hat{c}, c^*)$ that can be used to train specialized systems.

### 3.1 TRAINING THE DIFFUSION MODEL

The pre-trained components of CODEFUSION generate code from pure Gaussian noise. Because there is no natural language, we can remove the encoder. A denoiser $N$ removes the noise from $\mathbf{x}_t$ at timestep $t$ to obtain the denoised embeddings $\hat{x}_0 = N(\mathbf{x}_t, t)$. A decoder $D$ performs full self-attention over $\hat{x}_0$ to compute a decoded representation $D(\hat{x}_0)$. This allows each denoised token to be generated with information about other tokens, and improved the likelihood of generating

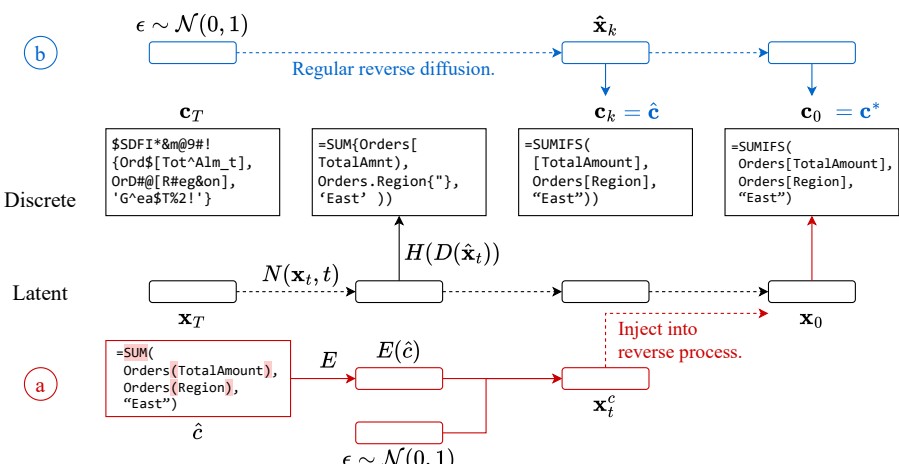

Figure 3: Using a pre-trained diffusion process (in black) to ⓐ repair broken code and ⓑ generate (broken, fixed) code pairs for training specialized approaches. ⓐ The broken code is embedded, noise is added for a timestep $t$, and the reverse process is resumed as usual, letting the reverse process fix the code. ⓑ The diffusion process produces intermediate (broken) code snippets $\hat{c}$ that can be paired with the final code $c^*$ to form a training example.

syntactically correct code (Singh et al., 2023a). Finally, the classification head $H$ computes $p(y \mid d_i)$ for each $d_i \in D(\hat{x}_0)$ to project decoded embeddings back to discrete tokens.

To train these components on a code snippet $\mathbf{c}$, an embedding layer $E$ first obtains the continuous representation $\mathbf{x}_0 = E(\mathbf{c})$. We sample $t \in [1, \ldots, T]$ and $\epsilon_t \sim \mathcal{N}(0, 1)$ and compute $\mathbf{x}_t$ from $\mathbf{x}_0$. The model is trained on

$$\mathcal{L} = \underbrace{\|N(\mathbf{x}_t, t) - \mathbf{x}_0\|}_{1} + \underbrace{\|D(\hat{x}_0) - E(\mathbf{c})\|}_{2} - \underbrace{\mathrm{ce}(\mathbf{c}, H(D(\hat{x}_0)))}_{3}$$

and consists of three parts that

1. minimize the error between the predicted noise $\hat{\epsilon}_t$ and the actual noise $\epsilon_t$ to train $N$,

2. minimize the error between the decoded embeddings $D(\hat{x}_0)$ and embedded code $E(\mathbf{c})$ to train $D$ and $L$, and

3. apply cross-entropy loss with respect to the ground truth code snippet $\mathbf{c}$ to train $H$.

This loss is taken from CODEFUSION (Singh et al., 2023a) and is an adaptation of the loss function used by GENIE (Lin et al., 2023).

## 3.2 DIFFUSION STEPS AS REPAIR OPERATORS

We exploit the Markov property of the reverse diffusion process to *inject* an embedded version of the noisy snippet into the reverse process. In other words, we can pick some $t$, generate $\epsilon \sim \mathcal{N}(0, 1)$ and compute $\mathbf{x}_t^{\hat{c}} = \sqrt{\bar{\alpha}}E(\hat{c}) + \sqrt{1 - \bar{\alpha}_t}\epsilon$ where $E$ is the embedding layer (that CodeFusion discards after training). The diffusion process then denoises $\mathbf{x}_t^{\hat{c}} \rightarrow \mathbf{x}_0^{\hat{c}}$ and we return $H(D(N(\mathbf{x}_0^{\hat{c}}, 0)))$.

Let $\mathbf{X}_t^{\hat{c}}[E]$ be the space of embedded representations $\mathbf{x}_t^{\hat{c}}$ obtained from $\hat{c}$ for all $\epsilon \sim \mathcal{N}(0, 1)$ at step $t$ (parametrized by $E$). Let $\mathbf{X}_t^{c^*}[N, D, H]$ be the space of embedded representations encountered at step $t$ in reverse diffusion processes starting from $\epsilon \sim \mathcal{N}(0, 1)$ that end up in $c^*$ (parametrized by $N$,

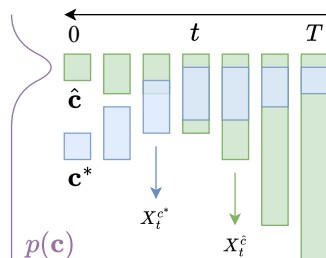

Figure 4: Overlap between $\mathbf{X}_t^{\hat{c}}$ (green) and $\mathbf{X}_t^{c^*}$ (blue) indicates that we can find some $t$ for which the embedding will project $\hat{c}$ into a trajectory that ends up in $c^*$.

$D$ and $H$). Our intuition is that there exists some $t$ for which these spaces have a significant overlap, and there are thus many values of $\epsilon$ that project $\hat{\mathbf{c}}$ into a trajectory to $\mathbf{c}^*$. If $t$ is too large, the probability of ending up there is small (too much noise). If $t$ is too small, it will never end up there (not enough noise). Figure 4 summarizes this.

### 3.3 DIFFUSION MODELS AS REPAIR GENERATORS

We exploit the seemingly discrete nature of later diffusion steps to generate synthetic repair data. Starting the reverse process from $\mathbf{x}_T \sim \mathcal{N}(0, 1)$ we build the chain $\hat{\mathbf{x}}_T \to \hat{\mathbf{x}}_0$ and decode each snippet into $\mathbf{c}_T \to \mathbf{c}_0$. We can then select any $(\mathbf{c}_t, \mathbf{c}_0)$ as a training pair if $\mathbf{c}_t \neq \mathbf{c}_0$.

In previous work, mistakes are introduced in the discrete token space, by implementing specialized functions that imitate human errors (Yasunaga & Liang, 2020; Joshi et al., 2024) and optionally training a neural network to imitate those (Yasunaga & Liang, 2021). Our aim is to show that the space of discrete representations encountered during the reverse diffusion process shares enough similarities to the discrete errors that humans make to be useful for last-mile repair.

## 4 EXPERIMENTS

We evaluate both how the diffusion process acts as a repair operator, how the generated data can be used for supervised repair training, and provide additional insights in how the diffusion generates (and repairs) code.

### 4.1 EXPERIMENTAL SETUP

**Benchmarks**  We evaluate our approach on three different benchmarks that span different types of code (formulas, code, commands).

1. **Excel** (Bavishi et al., 2022) is a benchmark of 200 broken formulas mined from a public Excel help forum[2].

2. **PowerShell** (Joshi et al., 2023) is a repair benchmark for 208 PowerShell commands collected from StackExchange[3] by comparing commends in the question with those in accepted answers.

3. **Python** (Yasunaga & Liang, 2021) is a code repair benchmark collected from GitHub. We evaluate on a random sample of 200 syntactically invalid Python code snippets. These do not have a ground truth repair, hence, we employ the same evaluation metric described in the BIFI paper using (1) syntactic validity and (2) token edit distance $< 5$.

**Pre-training data**  Collecting snippets of code for unsupervised approaches is significantly easier than finding data for repair.

1. For **Python**, we use a collection of code snippets for simple tasks from StackOverflow[4]. The corpus has 130K snippets with an average token length of 79.4 tokens.

2. For **Excel**, we use a corpus of 1.8 million workbooks (Singh et al., 2022), and sample 200K workbooks and collect all formulas present in them to generate 108K unique formulas with an average length of 35.8 tokens.

3. For **PowerShell**, we mine PowerShell commands from StackOverflow and other online forums ourselves. The corpus has 110K samples with an average length of 24.9 characters.

**Metrics**  When available, we use execution match—comparing the output of executing the repaired code with an expected output—which allows for semantically different but functionally equivalent code snippets. To further analyze the syntactic closeness of the repairs to the original code, we also report sketch match, which is implemented as the exact string match of

---

[2]www.mrexcel.com

[3]www.stackexchange.com

[4]www.stackoverflow.com

| Denoiser | Decoder | #P | Noise | Python | | PowerShell | | Excel | |
|---|---|---|---|---|---|---|---|---|---|
| | | | | Sketch | Execute | Sketch | Execute | Sketch | Execute |
| CF | CF | 45M | any% | 65.3 | 68.1 | 14.3 | 21.2 | 62.3 | 63.4 |
| | | | best% | 60.4 | 62.0 | 11.0 | 17.4 | 56.2 | 58.9 |
| | | | vote% | 61.2 | 62.4 | 11.7 | 18.2 | 57.1 | 59.1 |
| Unet | Clamp | 15M | any% | 19.4 | 20.3 | 2.1 | 3.5 | 17.8 | 18.4 |
| | | | best% | 19.1 | 20.1 | 1.8 | 3.3 | 16.7 | 17.3 |
| | | | vote% | 19.1 | 20.2 | 1.8 | 3.3 | 16.9 | 17.5 |
| | Decoder | 15M | any% | 34.2 | 35.4 | 3.4 | 5.6 | 26.5 | 27.2 |
| | | | best% | 31.3 | 32.6 | 3.1 | 5.2 | 23.4 | 24.5 |
| | | | vote% | 31.5 | 33.0 | 3.1 | 5.3 | 22.8 | 23.1 |
| Transformer | Clamp | 30M | any% | 55.2 | 56.2 | 8.7 | 13.5 | 50.2 | 51.1 |
| | | | best% | 51.1 | 52.5 | 7.4 | 11.8 | 46.5 | 47.3 |
| | | | vote% | 51.9 | 52.8 | 7.5 | 12.1 | 47.3 | 48.2 |
| | Decoder | 45M | any% | 64.7 | 66.9 | 14.2 | 21.2 | 60.5 | 61.1 |
| | | | best% | 58.9 | 60.2 | 10.7 | 16.8 | 54.2 | 56.8 |
| | | | vote% | 60.1 | 61.3 | 11.4 | 17.9 | 54.8 | 57.8 |

Table 1: Repair results for different diffusion architectures (#P is number of parameters and CF is CodeFusion). We report sketch and execution match for all languages at different noise settings: (1) **any%** denotes any noise level was able to satisfy for each sample; (2) **best%** denotes picking the best noise level across all samples; (3) **vote%** denotes voting among different noise levels and considering the most frequent code generation as the result. We find that all models have capacity to perform code repair.

code with constants (strings, numbers, cell references) anonymized. For example, a noisy code snippet `SUM(A1:A10)/COUNT(A1:A10)` and its diffusion repaired snippet `AVERAGE(A1:A10)` match in execution but not in their sketches (`SUM(:)/COUNT(:)` and `AVERAGE(:)`). On the other hand, `SUM(A1:A5)/COUNT(A1:A5)` may not be an execution match but is a sketch match.

**Models**   We implement the same architecture and pre-training as CodeFusion (Singh et al., 2023a). The **embedding** ($E$) has a dimension of 512. The **denoiser** ($D$) is a transformer encoder (Vaswani et al., 2017) with 10 transformer blocks. The **decoder** ($D$) is a block with 6 transformer decoder layers. The **classification head** ($H$) is a single fully connected layer. Additionally, we try the following ablated variations.

- We replace the denoiser with U-Net (Ronneberger et al., 2015)—a common denoiser for image diffusion—with standard hyper-parameters (Rombach et al., 2022b).

- We replace the decoder and classification head with a clamping approach that rounds each denoised token to the closest embedded token (Li et al., 2022) and adapt the loss function to only incorporate the denoiser (1) and classification (3) components.

- We remove the paragraph denoising objective, which instead of adding noise to all tokens, only adds noise to language keywords (like `SUM` for Excel, or `map` for Python).

## 4.2 DIFFUSION FOR CODE REPAIR

We evaluate a pre-trained diffusion model on last-mile repair. Table 1 contains the execution and sketch match, pooled across noise levels using three strategies, for different diffusion architectures. In **any%**, any noise level was able to correctly repair the code for each sample, indicating the promise of diffusion for repair. In **best%**, we pick the best global noise level for each benchmark set, which are indicated in Figure 5. In **vote%**, we pick the repaired code that was obtained most often across noise levels (using exact string match).

Our findings show that all variations of diffusion models have repair capabilities, with transformer-based architectures performing ($\sim$ two to four times) better. The decoder and classification head,

which are aimed at improving the syntactic validity of code, remain important components. Vote-pooling across noise levels is *slightly* more effective than the *free lunch* of an optimal noise level.

Figure 5 shows how the execution match evolves in function of the noise level (increments of 10%) and marks the "optimal" noise level (based on average + one standard deviation). Last-mile repairs are typically small, causing all lower noise levels to work. For larger noise levels, we see a decline in performance, as the model makes too many changes to the code.

Additionally, we examine how error complexity correlates with the noise levels required for repair. Figure 6 shows an area plot of maximum and minimum noise levels where the correct code is generated at least once with increasing complexity, computed as normalized edit distance for the repair tasks. The results suggest the acceptable noise band varies based on the complexity where an earlier injection is preferred for more complex tasks as these require more iterations to repair. Furthermore, across languages, we see that Excel has a much wider band as it requires fewer edits while for Python and PowerShell more edits are required for the repair.

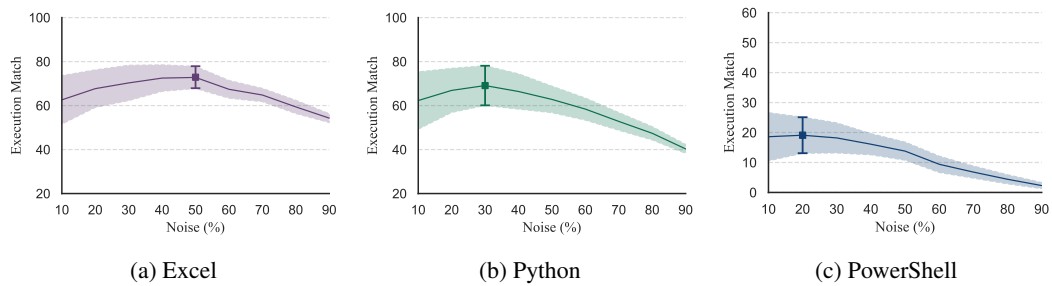

|          (a) Excel          |          (b) Python          |          (c) PowerShell          |

Figure 5: The evolution of execution match for increasing noise levels added to the noisy snippet ($\hat{c}$). The optimal noise level is marked. We find that for simpler languages like formulas, injecting later helps while for more complex languages like Python and PowerShell, injecting earlier gives the model more time to repair to the correct code.

To put our results in perspective, Table 2 compares the pass@1 rate of the pre-trained diffusion model with existing approaches. For LaMirage (Bavishi et al., 2022) and BIFI (Yasunaga & Liang, 2021) we report the numbers from their respective papers. For the other approaches, we re-implement them, with the Codex (Chen et al., 2021) and GPT-4o results based on the RING prompt without compiler feedback. We note that these very powerful models (GPT-4o), specific repair systems (BIFI and LaMirage) and using additional context (RING) still perform better. Still, outperforming the Codex model on Python (+8%) and PowerShell (+11%) with a small (60M parameter) model that was not specifically trained for repair, is a remarkable result that indicates significant potential of applying diffusion to code repair.

| Approach | Type | Year | Python | | Excel | | PowerShell | |
|---|---|---|---|---|---|---|---|---|
| | | | Sketch | Exec. | Sketch | Exec. | Sketch | Exec. |
| Codex | Prompt | 2021 | 0.56 | 0.60 | 0.65 | 0.67 | 0.08 | 0.10 |
| RING | Prompt | 2022 | 0.78 | 0.82 | 0.68 | 0.74 | 0.15 | 0.18 |
| GPT-4o | Prompt | 2024 | 0.81 | 0.84 | 0.68 | 0.75 | 0.15 | 0.24 |
| LaMirage | Fine-tuned | 2022 | 0.67 | 0.71 | 0.69 | 0.72 | – | – |
| BIFI | Fine-tuned | 2021 | 0.72 | 0.76 | – | – | – | – |
| CodeFusion | Pre-trained | – | 0.65 | 0.68 | 0.62 | 0.63 | 0.14 | 0.21 |

Table 2: Comparison performance of CodeFusion with state-of-the-art last-mile repair approaches.

A major advantage of CodeFusion is its ability to generate diverse outputs, as it is conditioned on noise. Figure 3 shows the pass@1, pass@3 and pass@5 rates for diffusion, GPT-4o and RING. CodeFusion sees the biggest jump in performance ($\pm$ 5%) across all languages, even performing better than GPT-4o on the (most difficult) PowerShell benchmark. This reinforces the potential of diffusion for last-mile repair. Pooling over different noise vectors, execution feedback and larger diffusion models can leverage this potential even further, which we leave for future work.

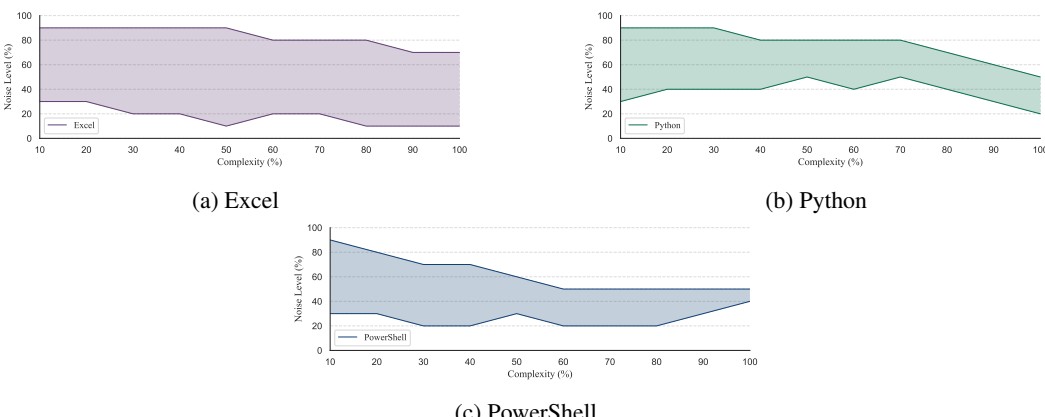

(a) Excel

(b) Python

(c) PowerShell

Figure 6: Noise range for which the correct code snippet is recovered for increasing differences between the broken and fixed code. We show the maximum and minimum noise for which a sample was repaired correctly. The band width is largest for Excel since it requires simpler and fewer modifications while PowerShell has a narrow band towards higher noise as it needs more iterations to repair.

| Approach | Python | | | Excel | | | PowerShell | | |
|---|---|---|---|---|---|---|---|---|---|
| | p@1 | p@3 | p@5 | p@1 | p@3 | p@5 | p@1 | p@3 | p@5 |
| CodeFusion | 68.1 | 70.5 | 72.4 | 63.4 | 65.8 | 68.2 | 21.2 | 23.1 | 26.4 |
| GPT-4o | 81.2 | 81.7 | 82.1 | 75.3 | 75.6 | 75.7 | 23.9 | 24.1 | 24.2 |
| RING | 82.4 | 82.6 | 82.9 | 73.8 | 74.2 | 74.5 | 18.0 | 18.0 | 18.2 |

Table 3: Pass@$k$ rates for repair for the best diffusion model adapted for repair (CodeFusion). The performance for CodeFusion increases the most when increasing $k$, as it is able to generate diverse repairs due to its noise condition.

## 4.3 DIFFUSION FOR SYNTHETIC DATA GENERATION

We evaluate the pre-trained diffusion model (CodeFusion architecture) on generating training data for supervised approaches. We uniformly sample $t$ and select $(\mathbf{c}_t, \mathbf{c}_0)$ from the diffusion process. We then fine-tune several code generation models on this dataset and evaluate their performance on a repair benchmark containing real human errors. We sample 20K training points.

As baselines, we consider generators from existing work, as well as generating data with a large language model (GPT-4o). For Python, we use the popular BIFI (Yasunaga & Liang, 2021) model, which learns to break code based on a set of manually curated repair operators. For Excel, we use the 17 operators used to fine-tune FLAME (Joshi et al., 2024) on last-mile formula repair. The prompt for GPT-4o is a few-shot, chain-of-thought prompt where we instruct the model to break a formula according to mistakes that a human would make. We use two versions: (1) using the error categories from BIFI to mimic common human errors and (2) not providing guidance to promote diversity in the mistakes. We have included a template of this prompt in Appendix A.1.

Table 4 shows the performance of different data generation techniques across various models: CodeT5+ (2B) (Wang et al., 2023), Phi-3.5-mini-instruct (3.8B) (Abdin et al., 2024) and Mistral-7B-instruct-v0.3 (7B) (Jiang et al., 2023). Our results show that models trained on diffusion-generated consistently perform better than or on part with even the specialized approaches, across all models. Similar to the repair performance, a significant contributor is the diversity in the generated data, which is harder to control for GPT-4o.

We analyze properties of the generated data distributions for diffusion and GPT-4o in Figure 5. We show the average distance between token edits (localization), average n-gram similarity between randomly sampled data points (diversity), and average token edit distance between the noisy and correct code (complexity). Diffusion-generated data has more diversity, higher complexity, and

more global errors. The diffusion model generates both the code and the error from pure noise, whereas GPT-4o starts from provided code.

| Generator | Repair Model | Python | | PowerShell | | Excel formula | |
|---|---|---|---|---|---|---|---|
| | | Sketch | Exec. | Sketch | Exec. | Sketch | Exec. |
| Diffusion | CodeT5+ (2B) | **89.2** | **91.1** | 25.4 | **34.2** | 72.0 | **77.6** |
| | Phi-3.5-mini-instruct (3.8B) | 87.5 | 88.3 | **28.2** | 33.2 | 71.0 | 76.8 |
| | Mistral (7B) | 87.1 | 89.3 | 27.4 | **34.2** | **73.3** | 75.6 |
| GPT-4o | CodeT5+ (2B) | 87.6 | 88.2 | 23.4 | 28.1 | 69.2 | 72.1 |
| | Phi-3.5-mini (3.8B) | 84.2 | 86.9 | 21.0 | 27.3 | 70.1 | 74.3 |
| | Mistral (7B) | 85.4 | 87.7 | 24.5 | 29.4 | 69.3 | 70.0 |
| Syntactic | CodeT5+ (2B) | 85.4 | 87.3 | – | – | 70.1 | 72.4 |
| | Phi-3.5-mini (3.8B) | 84.2 | 85.3 | – | – | 72.4 | **77.6** |
| | Mistral (7B) | 86.0 | 89.3 | – | – | 71.2 | 73.5 |

Table 4: Results on fine-tuning different language models on the synthetic repair data generated by diffusion, GPT-4o and Syntactic systems (BIFI (Yasunaga & Liang, 2021) for Python and `flame` (Joshi et al., 2024) for Excel). We see that Diffusion generated data consistently performs better than language model and syntactic systems.

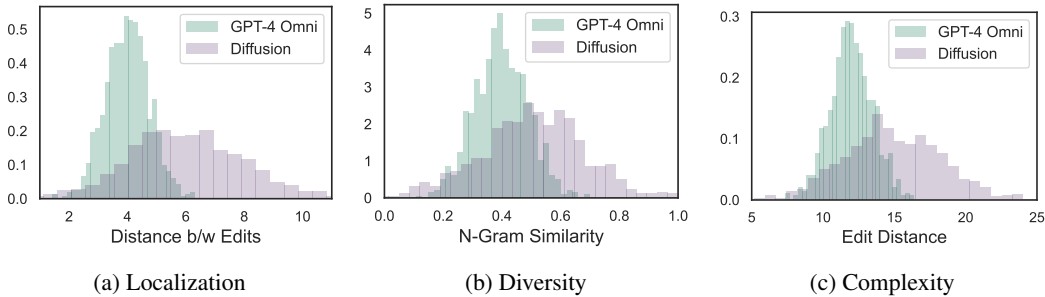

(a) Localization          (b) Diversity          (c) Complexity

Figure 7: Figure showing the trends in the diffusion (purple) and GPT-4 (green) generated repair data. We show (a) Localization—average distance between edits; (b) Diversity—average n-gram diversity in generated correct code; and (c) Complexity—edit distance of the repair.

Finally, Figure 8 shows the overlap between Excel benchmarks solved using the CodeT5+ model for different sources of synthetic data. Using diffusion data solves all cases that are solved by the synthetic data, which required manual analysis of human errors to manually implement 17 noise operators. Bigger mistakes, like completely missing an argument spanning multiple tokens, occur more in the diffusion data. An extra parenthesis does not occur as much in the diffusion data, as the pre-trained models quickly learns this structure, and is an explicit instruction in the GPT-4o prompt.

## 5 RELATED WORK

**Diffusion models for text and code** Diffusion models have shown their ability to gradually refine noisy data into realistic outputs through a denoising process (Sohl-Dickstein et al., 2015). They were originally popularized to generate photo-realistic images (Ho et al., 2020; Rombach et al., 2022a) and later applied to other high-dimensional data generation, like audio (Kong et al., 2021) and video (Ho et al., 2022) synthesis. Diffusion has also been adapted for to discrete domains like text (Li et al., 2022; Lin et al., 2023) and code (Singh et al., 2023a) where the ability to look at the whole previous generation has benefits over auto-regressive generation. Two approaches are embedding discrete tokens into a continuous space where the diffusion takes place and then decoding Li et al. (2022) or directly performing diffusion in the discrete space through a transition matrix He et al. (2023). In this paper, we use the former approach to explore the latent code repair capabilities of these models.

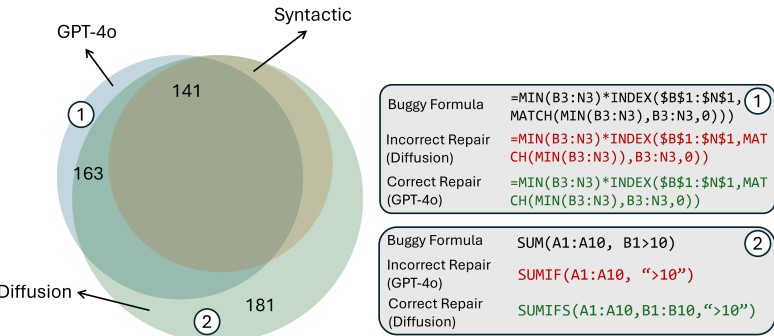

Figure 8: Venn diagram of benchmarks solved correctly for models trained on synthetic datasets generated from different sources. Diffusion-generated data supersedes all syntactic cases. The example shows cases where diffusion data trained model is able to repair a task while GPT-4o data trained model cannot and vice versa.

**Code repair** Automated code repair (Zhang et al., 2023) has long been a key challenge in software engineering, with early approaches using heuristic searches (Qi et al., 2014) and program synthesis (Nguyen et al., 2013; Bavishi et al., 2022). More recently, transformer-based systems have been shown adept at learning to repair code (Berabi et al., 2021; Yasunaga & Liang, 2021; Tufano et al., 2019). A major limitation of training a repair model is the requirement for large quantities of data. That is not true anymore for large language models, which are adept at repair code and can take in additional context like error messages (Joshi et al., 2023). They are expensive to deploy, however, and it is much harder to steer them to remain close to the original code snippet.

**Human error simulation** In order to leverage the corpora of unsupervised data, previous works have explored generating synthetic data using static rules (Joshi et al., 2024; Gupta et al., 2017; Hellendoorn et al., 2019) or learning to break programs in a *natural* way (Yasunaga & Liang, 2020; 2021). These approaches are limited to the encoded rules and the quality of the learned code breaker—which depends on the training data—and thus suffer from out-of-domain generalization. In this work we aim to generalize the synthetic data to the long tail of out of domain cases.

## 6 CONCLUSION AND LIMITATIONS

In this paper, we explored the potential of applying pre-trained code diffusion to the problem of last-mile repair. These diffusion models iteratively denoise a latent representation of code and the discrete decoding of intermediate steps resemble last-mile programming errors. Our experiments show that injecting actual broken code into this process can cause the diffusion process to repair the code, and that sampling these intermediate step yields data that can be used to fine-tune last-mile repair models. In its current state, using diffusion models to generate synthetic training data shows the most promise.

Diffusion for code has only been applied to shorter snippets with smaller models, on relatively small datasets. Since there is no additional context, like error messages or test cases, the model might not capture some of the semantics of the broken snippet. Scaling up the model, context and data should further improve the potential of these models. Our findings consider the diffusion model as-is: controlled decoding (Li et al., 2022) can help in remaining close to the source snippet. Like other work on diffusion for text (Li et al., 2022), we note that inference is slower than auto-regressive models. We can leverage work on both efficient diffusion models and efficient transformers to speed up the model.

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

# A    APPENDIX

## A.1    SYNTHETIC DATA GENERATION USING GPT-4O

We use a few-shot, chain-of-thought prompt to generate data. The prompt is inspired from previous work (Singh et al., 2023b) which also generates synthetic data. The prompt template is shown below.

```
You are an expert in {{Language}} and generating errors in them.

<Task>
- You are given a code snippet and you need to introduce errors in it.
- The errors need to be human like.
- The errors can be either syntactic or semantic.
- For succesful completion of this task, you need to perform three steps:
1. Explanation: explain what the code snippet is doing.
2. Error Reasoning: list down what are potential errors
   humans might make in writing this code.
3. Buggy Code Generation: generate the buggy version of the code.

<Input>
You are given the correct code snippet in a code block.

<Output>
You need to generate the three steps for this task.
For example,
"""
1. Explanation:
<explanation of code>

2. Error Reasoning:
- <potential error 1>
- <potential error 2>
...

3. Buggy Code Generation:
```
<buggy version of code>
```
"""

<Examples>
{{Examples}}

<Input Code>
{{Code}}

<Steps>
```

## A.2    DIFFUSION REPAIR PROCESS SAMPLE

