# OpenReview forum: "Code diffusion models are continuous human noise operators"
_ICLR.cc/2025/Conference — Submitted to ICLR 2025_

### Official Review · Reviewer_VztF · 2024-11-01

**Soundness:** 3
**Presentation:** 2
**Contribution:** 3
**Rating:** 5
**Confidence:** 4

**Summary:**

This paper uses pretrained code diffusion model to fix erroneous code snippets and to obtain correct-erroneous code snippet pairs for subsequent finetuning of code models on code repair tasks. Paper reimplements CodeFusion model with additional variants and evaluates the models on code repair tasks for 3 programming languages. Next paper demonstrates the high quality of generated code pairs by comparing to using syntactic systems and gpt-4o to generate the data. Additionally this paper performs an analysis study of how diffusion model performs the code fixes.

**Strengths:**

The paper tackles an important problem of automatically fixing erroneous code snippets. The paper propose a novel application of code diffusion models to the task of code repair. They propose to directly use diffusion model reverse process to fix the erroneous code snippets and to generate training data for other code repair models, which is to my understanding a novel approach, and using diffusion models for the code repair task seems like a reasonable idea.

Authors experiments on three languages: Python, PowerShell, Excel.

Analysis from this paper is interesting and I believe will be valuable for code community.

**Weaknesses:**

- While the paper focuses on the code repairing, it is hard to understand the assumptions of this work, which types of code errors (syntax errors, compilation errors, variable misuse, interface errors, etc. ) the diffusion model is expected to fix?
- No direct comparison to any other code repair literature makes it hard to evaluate the usability of this approach and the quality of code repair.
- No details described for the finetuning of the models on the generated training data (4.3) hinders reproducibility.
- The assumptions on human cognition patterns (how humans treat code errors) are not supported by any evidence or link to the literature, which also makes the title misleading.
- The paper contains many typos and the writing is not always precise (see questions)

**Questions:**

- as far as I understand, to repair code snippet, this work first adds a noise to it and then apply several reverse diffusion steps to reconstruct it? Why do you add additional noise to the erroneous snippet? My intuition would be that a code sample with errors is already "noisy".
- is $\epsilon$ noise in line 178, 185 the single scalar or it should be a vector/matrix? From the writing it seems that it is a scalar, but I think it should be a matrix.
- how any% best% pooled% are exactly computed? Does it mean that we sample different $\epsilon$ or different noise powers?
- How do you ensure that data from pre-training does not intersect with data used during evaluation? Do you perform any deduplication?
- Can you point to the references regarding the comparison to humans: "human error patterns", "closely mirrors how humans tackle complex problems" discussed in this work?

Typos and unclear writing:
- 029: from the distribution -> from a distribution?
- 042: not sure why "thus" is here
- 074: define "continuous human noise operator"
- define "last-mile" error
- 081: remove brackets around 56.4-68.2%
- 090 Diffusion models: typo in paragraph name;
- clarify the family of diffusion models used in this work?
- 048: what is the notation used here, and what are the "specifications", what is $d$? $T$ here is not the same as diffusion time step I assume?
- 170: Is the loss the same as in GENIE (Lin et al.,2023)? in 1. loss part should $\epsilon_t, \hat{\epsilon}_t$ be part of the loss definition?. In 2. what is $D_s$ in loss definition?
- Figure 3: what are the bars here and how to interpret the intersection between them? I am not sure I understood this figure.
- 207: "different styles of code", I am not sure the styles is the right word here, maybe you mean programming languages?
- 251: the classification head here is a linear transformation?
- 255: which code repair task? Citation?
- Table 1: what is Clamp vs Decoder? Maybe add citations here to Unet and Transformer (or point to where you formulate these models in detail?)
- Figure 4: I am not sure I understand the conclusion, what is Noise (%) here for X axis on the plots?
- 314: add citations for syntactic generators and gpt-4o.


Links: Genie, Lin et al. 2023: https://arxiv.org/pdf/2212.11685

---

> ### Author Response · Authors · 2024-11-23
> **Rebuttal**
>
> **Question: Which types of code errors the diffusion model is expected to fix?**
>
> We focus on syntax errors and will make this assumption more explicit.
>
> \
> **Question: No direct comparison to any other code repair literature makes it hard to evaluate the usability of this approach and the quality of code repair**
>
> We do not claim that, in its current state, diffusion should be used as a practical repair approach. Our paper makes fundamental observations about repair behavior of a reverse diffusion process for code, showing that (1) it can repair syntax errors and (2) it can be used to generate training data for a specialized model.
>
> To make these insights stronger and provide insights into the complexity of the benchmarks, we can report the performance of existing repair approaches on these benchmarks. For example, the RING [1] paper that introduced the PowerShell benchmarks, reports a performance of 10% in pass@1 and 27% in pass@50 using the 175B parameter Codex model. We show a repair performance of 18% using pooled diffusion and 34.2% on fine-tuning a codet5-small (60M parameters) on synthetic data.
>
> We include a comparison to baselines below for reference, we will include this in the paper.
>
> | System   | Python |           | Excel  |           | Powershell |           |
> |----------|--------|-----------|--------|-----------|------------|-----------|
> |          | Sketch | Execution | Sketch | Execution | Sketch     | Execution |
> | RING     | 0.78   | 0.82      | 0.68   | 0.74      | 0.15       | 0.18      |
> | GPT-4o   | 0.81   | 0.84      | 0.68   | 0.75      | 0.15       | 0.24      |
> | LaMirage | 0.67   | 0.71      | 0.69   | 0.72      | -          | -         |
> | BiFi     | 0.72   | 0.76      | -      | -         | -          | -         |
> | Codex    | 0.56   | 0.6       | 0.65   | 0.67      | 0.08       | 0.1       |
>
> \
> **Question-3: No details described for the finetuning of the models on the generated training data (4.3) hinders reproducibility.**
>
> We will add all the details to the paper. In a high level, we synthetically generate buggy and correct code pairs and then train the model to generate the correct code from the broken code. We employ the standard finetuning described by the authors.
>
> Furthermore, we analyzed the tasks solved by different synthetic data techniques and found that there is a very high overlap between syntactic and GPT-4o and the cases solved by diffusion models are different from these. To give an example, the buggy formula "SUM(A1:A10, B1>10)" is repaired as "SUMIF(A1:A10, ">10")" which is incorrect as it doesnt compare with the B column. The model trained on diffusion generated data, is able to generate the correct repair "SUMIFS(A1:A10, B1:B10, ">10")".
>
> We also include the Pass@K results and will add these to the paper:
>
> | Dataset    | Pass@1 | Pass@3 | Pass@5 |
> |------------|--------|--------|--------|
> | Python     | 68.1   | 70.5   | 72.4   |
> | Excel      | 63.4   | 65.8   | 68.2   |
> | Powershell | 21.2   | 23.1   | 26.4   |
>
> \
> **Question-4: The assumptions on human cognition patterns (how humans treat code errors) are not supported by any evidence or link to the literature, which also makes the title misleading.**
>
> Whereas our intuition was that intermediate code in the reverse diffusion process must have some resemblance to human errors if it can repair them, we indeed do not conclusively show that diffusion models follow human cognition pattern. We will rephrase the abstract, introduction and Section 4.4 to reduce the emphasis on human cognition and focus on the (empirical) overlap in distributions between human errors and intermediate code in the reverse diffusion process.
>
> Previous work has noted human repair patterns in code [1]. We will cite them and clarify the connection and scope of the work.
>
> [1] Dominik Huber and Matteo Paltenghi and Michael Pradel, Where to Look When Repairing Code? Comparing the Attention of Neural Models and Developers, 2023, https://arxiv.org/abs/2305.07287.

---

> > ### Author Response · Authors · 2024-11-23
> > **Rebuttal Contd.**
> >
> > **Question-5: As far as I understand, to repair code snippet, this work first adds a noise to it and then apply several reverse diffusion steps to reconstruct it? Why do you add additional noise to the erroneous snippet? My intuition would be that a code sample with errors is already "noisy"**
> >
> > The broken code sample is “noisy in a denoised state.” In other words, there is noise in the token (discrete) space. A diffusion model is trained to iteratively remove noise from latent representation x_t. When we take such an intermediate state x_t, it will thus have two levels of noise: the (in our case Gaussian) noise in the latent space that the denoiser learns to remove, and the noise in the discrete space that is still left because the denoiser is not perfect. The denoiser model is trained to remove the first type of noise in each iteration to get to a state (around time = 0) where the second type of noise will be (implicitly) removed as well. So, to use that implicit repair, we “emulate” the first type of noise (for some given step t) and act as if the diffusion model obtained that noised vector as part of its reverse process.
> >
> > \
> > **Question-6: Is ϵ noise in line 178, 185 the single scalar or it should be a vector/matrix? From the writing it seems that it is a scalar, but I think it should be a matrix.**
> >
> > Indeed, the noise ϵ will be a matrix of the same shape as the latent code representation. We will bold-face the epsilon to make this distinction clear, as done in literature [1].
> > [1] https://arxiv.org/abs/2006.11239
> >
> > \
> > **Question-7: how any% best% pooled% are exactly computed? Does it mean that we sample different ϵ or different noise powers?**
> >
> > They are computed by sampling different noise powers (0 to 100 with increments of 5). The results are averaged over N different samples of ϵ.  Pooling over different ϵ and noise powers is an interesting approach to potentially further improve performance.
> >
> > We include results for pooling over different ϵ and noise powers and will add in the paper.
> >
> > | Noise  | Python |           | Powershell |           | Excel Formula |           |
> > |--------|--------|-----------|------------|-----------|---------------|-----------|
> > |        | Sketch | Execution | Sketch     | Execution | Sketch        | Execution |
> > | any %  | 65.5   | 68.4      | 14.5       | 21.3      | 62.5          | 63.7      |
> > | best % | 60.7   | 62.4      | 11.3       | 56.7      | 56.7          | 59.4      |
> > | pooled | 61.7   | 63.1      | 12.1       | 18.6      | 57.3          | 59.4      |
> >
> > \
> > **Question-8: How do you ensure that data from pre-training does not intersect with data used during evaluation? Do you perform any deduplication?**
> >
> > The data used for evaluation (“Benchmarks” in 4.1) is completely disjoint from the pretraining data used to train the diffusion models. The synthetic data generated by the diffusion models (Section 4.2) could theoretically generate a data point very close to a test sample. Even though the chance of this happening is close to zero, we de-duplicated this dataset with the test set (removing 0 cases). We will clarify this more in the paper.
> >
> > \
> > **Question-9: Can you point to the references regarding the comparison to humans: "human error patterns", "closely mirrors how humans tackle complex problems" discussed in this work?**
> >
> > As answered above, these statements were not properly worded, and we will ensure that any comparison to humans purely refers to an overlap between the distribution of syntactic errors that humans make, and the distribution errors encountered as intermediate steps in the reverse diffusion process.
> > Additionally, we will add more related work on human cognition in repair [1] and properly address the connection with our work.
> > [1] Dominik Huber and Matteo Paltenghi and Michael Pradel, Where to Look When Repairing Code? Comparing the Attention of Neural Models and Developers, 2023, https://arxiv.org/abs/2305.07287.
> >
> > \
> > **Question-10: Typos and unclear writing.**
> >
> > Thank you for the detailed analysis and addressing these issues. We will resolve all of them, or provide an explanation here:
> >
> > - **048: what is the notation used here, and what are the "specifications", what is d? There is not the same as diffusion time step I assume?**
> >
> > The specification is what a user had in mind; it can be anything. We have updated the definition to have less notation.
> >
> > - **Is the loss the same as in GENIE (Lin et al.,2023)?**
> >
> > The loss is the same as in CodeFusion, which is indeed based on based on GENIE. We have clarified.

---

> ### Comment · Reviewer_VztF · 2024-11-24
> **Thank you for your response**
>
> Dear authors, thank you for the answers, they resolve part of my concerns and raised my score. I still don't understand some things, could you help me clarify?
>
> > Question-7: how any% best% pooled% are exactly computed?
>
> pooled – picking the sample using maximum voting across different noise
> levels. I still don't understand how this is computed.
>
> Was the manuscript updated already / do you plan to add new things before the end of rebuttle?

---

> > ### Author Response · Authors · 2024-11-24
> >
> > Thank you for continuing the discussion.
> >
> > For each time-step t, we can get a repaired formula. In our experiments, we set t as a percentage of the total time-steps. For example, for 1200 time-steps, 10% is t = 120, 20% is t = 240, and so on. We get 9 repaired formulas in total.
> >
> > Any% means that we see if there exists any noise level that yields a correct repair. The challenge is then figuring out which is the correct repair, and this experiment purely serves to evaluate how promising the idea is. Best% and Pooled% are then two ways of picking one repair from all those different options.
> > - Best% means that we use the globally best time-step (within each domain). Figure 4 shows these noise levels. For example, for Excel, we always take the 50% noise level. This only requires that exact noise level to be computed.
> > - Pooled% means that we take the repaired code that was returned most often across these 9 samples, as it does happen that different noise levels yield exactly the same output.
> >
> >
> > So far, we have made some improvements to the presentation in the PDF. On Monday (25th) we will submit a new revision that also includes the additional results requested by reviewers, as well as some additional details on the experiments. We are also performing more experiments on pooling and variance across different noise vectors, which can also affect the result.

---

> > > ### Author Response · Authors · 2024-12-02
> > > **Rebuttal**
> > >
> > > We hope we were able to answer the concerns raised. Since today is the last day for submitting questions regarding our paper, we wanted to check in if there are any further clarifications. We appreciate your time, and the valuable feedback provided, which was valuables to improving our work.
> > >
> > > We have updated the manuscript to address the concerns raised by reviewers. As a brief summary, we have significantly enhanced the narrative of our paper, emphasizing the applications of diffusion for last-mile repair. We have made a range of improvements, including revisiting our claims, adding results for comparison, and including new visualizations to support our findings. The general comment describes the changes in greater detail
> > >
> > > If you have any further questions or require clarification on any aspect of the paper, we would be more than happy to address them before the deadline.

---

### Official Review · Reviewer_P1bD · 2024-11-02

**Soundness:** 3
**Presentation:** 3
**Contribution:** 3
**Rating:** 8
**Confidence:** 5

**Summary:**

This work analyzes whether (part of) the denoising process in code diffusion models mimics patterns seen in real-world program repair. It provides a range of evidence for this. It shows that diffusion models can be used to repair faulty programs by injecting noise into its embedding at the appropriate timestep in the denoising process and denoising as usual. It shows that buggy programs generated by the denoising process (before generating the correct solution) make for diverse and effective training data for program repair. And it provides a range of insights into why this works, mostly related to the way outputs of the diffusion process change over steps.

**Strengths:**

This work provides substantial evidence to back up an intuitive assumption, viz. that diffusion process mimic program repair to some extent. This involves reasonably significant contributions: both the method for repairing programs by introducing noise into their embeddings, and the method for generating large scale training data are noteworthy and effective. To the best of my knowledge, these are novel ideas, in that no prior work has provided empirical evidence for them. The results are well-presented and reasonably easy to follow.

**Weaknesses:**

The methodology, especially, could use some cleaning up. The paper uses quite a few one-letter abbreviations, some of them overloaded, and others not/inadequately explained. I list a few examples below, as well some (minor) incorrect statements.

The motivation doesn't fully connect to the results. The rationale given in the abstract and introduction is that diffusion operations may well resemble _human_ repair actions and suggests the work will investigate if denoising steps are "representative" of human repair steps. The work does not provide much proof for this conjecture. What it shows is that diffusion can repair broken programs and can be used to train program repair engines, but that is not the same as mimicking human repairs. Diffusion models learn to generate real programs from a wide variety of noisy ones. Some of those will end up resembling mistakes a human might make while others may look wildly different. At best, they train on a superset of human-like repair actions, though depending on the decoder, they may well miss some key human behaviors too. Proving that the repair actions are "human noise operators" (L74) would require a pair-wise comparison between a large dataset of human repairs and of diffusion repairs, which this work does not include. It does provide a few empirical observations of how diffusion-based repair operates (e.g. Fig. 7), but that is incomplete evidence for the premise. This doesn't subtract from the contributions of the work in terms of program repair effectiveness. I would just suggest toning down the discussion around human repair actions to note that there may be similarities but that this work does not prove that they are the same/a similar process (and in fact, that diffusion may involve many types of repairs that humans are very unlikely to encounter).

Minor Issues:

- L30: diffusion noise is not Gaussian by definition; that is just the most popular choice. Same at L92.
- L75: "changing" -> "changes"
- L148: the second occurrence of $c$ should also be $\hat{c}$
- L149: is $d$ a distance function, and if so, what distance does it measure?
- L157: $E$ is later also used to represent embeddings. Does it need to be used for the encoder if the encoder isn't used anyways?
- L182: is the inclusion of $N(...)$ here redundant, because it uses $x_0$ rather than $x_t$? The previous line already notes how $x_0$ is obtained.
- L237: nit: the sum & count formulas are incorrect; they need to sum to "A10", not "10".
- Fig. 5: the Python subplot, especially, suggests that more complex repairs require *less* noise (noise levels around 40%) than simpler ones, which contradicts the text. Does noise level refer to the step in which noise is introduced relative to the total number of steps? If so, the term is a bit confusing.
- L394: this sentence ends with comma, is it incomplete? It is unclear what 'more discrete than latent' means -- those concepts are not comparable.

**Questions:**

Given the drop-off between "best" and "any" in Tab. 1, what is the path to a practical approach to using diffusion for repair? Fig. 4 and 5 seem to imply that an initial guess of the distance between the broken and repaired code is often needed. It would help to know what the density is in the subplots in Fig. 5. Alternatively, do the results in Tab. 2 suggest that it is more effective to simply train auto-regressive models using diffusion-generated data, rather than to use diffusion to repair broken programs?

Please find suggestions for improving the work in the previous sections.

---

> ### Author Response · Authors · 2024-11-23
> **Rebuttal**
>
> **Question-1: The methodology, especially, could use some cleaning up. The paper uses quite a few one-letter, some of them overloaded**
>
> Thank you for the detailed analysis. We will use them to significantly improve the presentation of our paper.
>
> We include the Pass@k results below and will update the paper to show how multiple generations can improve performance.
>
>
> | Dataset    | Pass@1 | Pass@3 | Pass@5 |
> |------------|--------|--------|--------|
> | Python     | 68.1   | 70.5   | 72.4   |
> | Excel      | 63.4   | 65.8   | 68.2   |
> | Powershell | 21.2   | 23.1   | 26.4   |
>
> \
> We also add baselines to repair for comparison to Diffusion.
>
>
> | System   | Python |           | Excel  |           | Powershell |           |
> |----------|--------|-----------|--------|-----------|------------|-----------|
> |          | Sketch | Execution | Sketch | Execution | Sketch     | Execution |
> | RING     | 0.78   | 0.82      | 0.68   | 0.74      | 0.15       | 0.18      |
> | GPT-4o   | 0.81   | 0.84      | 0.68   | 0.75      | 0.15       | 0.24      |
> | LaMirage | 0.67   | 0.71      | 0.69   | 0.72      | -          | -         |
> | BiFi     | 0.72   | 0.76      | -      | -         | -          | -         |
> | Codex    | 0.56   | 0.6       | 0.65   | 0.67      | 0.08       | 0.1       |
>
> \
> **Question-2: The motivation doesn't fully connect to the results. The rationale given in the abstract and introduction is that diffusion operations may well resemble human repair actions and suggests the work will investigate if denoising steps are "representative" of human repair steps. The work does not provide much proof for this conjecture**
>
> We agree with the high-level observation that the diffusion process does not really resemble how a user repairs code. It is indeed the case that if any buggy code resembles broken code seen during the reverse diffusion process, the model will use its own strategy to solve that bug. This is shown in the results as well: a significant but not perfect proportion of the problems is solved. We will update our motivation and carefully revaluate any claims made, to make sure that they properly align with the results.
>
> \
> **Question-3: Given the drop-off between "best" and "any" in Tab. 1, what is the path to a practical approach to using diffusion for repair?**
>
> This is an excellent question! There are many possibilities, and turning our fundamental insights into a practical approach is definitely planned for future work. Some possibilities, which we will add to a future work section on the topic of going from insights to practice, are given here.
> - Pooling is one small step, which already slightly bridges performance. To continue from here, we can perform two levels of pooling, over different noise levels and over different noise vectors at each level.
> - We did not use any syntactic or execution-based filtering of the returned code snippets. If those are available, they allow to improve the pooling performance.
> - We did not analyze (yet) if failure cases are changing tokens outside of the actual bug. Instead of pooling over complete repairs, we can also pool over individual edits.
>
> We include results for pooling over different noise levels and \epsilon
>
> | Noise  | Python |           | Powershell |           | Excel Formula |           |
> |--------|--------|-----------|------------|-----------|---------------|-----------|
> |        | Sketch | Execution | Sketch     | Execution | Sketch        | Execution |
> | any %  | 65.5   | 68.4      | 14.5       | 21.3      | 62.5          | 63.7      |
> | best % | 60.7   | 62.4      | 11.3       | 56.7      | 56.7          | 59.4      |
> | pooled | 61.7   | 63.1      | 12.1       | 18.6      | 57.3          | 59.4      |
>
> \
> **Question-4: Fig. 4 and 5 seem to imply that an initial guess of the distance between the broken and repaired code is often needed. It would help to know what the density is in the subplots in Fig. 5.**
>
> We agree that since the optimal diffusion noise level depends on the broken formula distance which practically require sampling multiple noise levels. Whereas there is definitely some correlation, we actually expected it to be stronger. We will add the density to Figure 5, to show a more fine-grained insight.
>
> The aim of this paper is not to suggest a new repair method but rather explore latent repair abilities of diffusion models which no one has explored before.
>
> \
> **Question-5: Alternatively, do the results in Tab. 2 suggest that it is more effective to simply train auto-regressive models using diffusion-generated data, rather than to use diffusion to repair broken programs?**
>
> Without any improvements or deeper insights, that is indeed the more efficient option, both in terms of quality and inference speed. The goal of our research is to explore the potential of using a pre-trained diffusion model for syntactic code repair, and these are exactly the type of insights that we were hoping to get—we will add this more explicitly to the paper.

---

### Official Review · Reviewer_2pXc · 2024-11-04

**Soundness:** 3
**Presentation:** 3
**Contribution:** 3
**Rating:** 6
**Confidence:** 4

**Summary:**

This paper explores using an unconditional diffusion model for last-mile code repair under two different settings: (1) as a training technique to directly train code repair models and (2) as a training data technique to generate synthetic code repair examples. The authors evaluate their approach on benchmarks in the domains of Excel, PowerShell, and Python using various denoiser and decoder architectures. Their experiments show notable execution and sketch match performance on the Excel and Python benchmarks.

**Strengths:**

**Novel Application**: The paper applies the diffusion model—a technique initially popularized in continuous data like images—to a novel domain: code repair. The approach demonstrates the potential to adapt diffusion models for the discrete and syntactic requirements of code.

**Experimental Coverage**: The experiments are carried out across multiple programming languages and data types (Excel, PowerShell, Python), providing broad validation of the technique.

**Weaknesses:**

**Limited Significance of Performance Gains**: For some evaluation scenarios, such as synthetic data generation, the performance gains achieved using the diffusion-generated data appear to be only marginal. Given that the paper is presenting a new method, it would help to provide a more compelling differentiation from existing techniques.

**Incomplete Explanation on GPT-4o Prompting**: The authors do not provide adequate details on how GPT-4o was prompted for the evaluation. As performance in tasks like code repair can vary significantly depending on prompt quality, use of chain-of-thought reasoning, etc., it would be beneficial to have a complete description of the prompt structure either in the main text or in an appendix.

**Unexplored Trade-Offs**: The paper removes the natural language instruction component from the CodeFusion model, turning it into an unconditional diffusion model. However, the trade-offs involved in this design choice are not sufficiently explored. For example, one could imagine letting a large model such as GPT-4o generate an instruction describing the bug and then perform conditional diffusion on top.

**Questions:**

* What are the Y-axes representing in Figure 6? Providing this information will help in interpreting the trends depicted there.

* The authors claim that the diffusion process mirrors human debugging behavior. However, there are no concrete qualitative examples supporting this. Could the authors include additional examples (perhaps in the appendix) showing the sampled intermediate programs from the diffusion process?

* My understanding is that the diffusion is carried out in the latent space of N embedding vectors, where N is the number of tokens in the input program, and the decoder then takes in these denoised N vectors and generates M new tokens for the decoded program, where M is not necessarily same as N. Is this correct? If so, could the authors update the mathematical formulation and Figure 2 to make this clearer to the reader?

* It would be helpful to more explicitly differentiate this work from CodeFusion (Singh et al., 2023). Including a dedicated section or paragraph that clearly states the novel contributions of this paper compared to prior work would make the paper's contributions stand out more clearly.


* Will the authors make the code and data used in this work open-source?

---

> ### Author Response · Authors · 2024-11-23
> **Rebuttal**
>
> **Question-1: Limited Significance of Performance Gains: The performance gains achieved using the diffusion-generated data appear to be only marginal. Given that the paper is presenting a new method**
>
> We do not claim to present a new method. We show that a diffusion model can be used out-of-the box to generate training data for code syntax repair, because the distribution of “broken” code encountered during training has some overlap with the distribution of syntax mistakes that humans make. Thus, we do not train the diffusion model on repair and only show latent repair capabilities of the models.
>
> The presentation in Table 2 does not properly convey this point, by emphasizing the “best” result for each metric. We will update this in the paper. On Excel formulas, for example, the “syntactic” mistakes required designing 17 operators that break formulas. Getting the inspiration required looking at hundreds of mistakes, and implementing these (potentially) required a lot of formula parsing.
>
> Our approach just requires a corpus of formulas and nothing else and improves by +5.2% (codet5-small) and +2.1% (mistral-7B) on the two models that were not trained by Microsoft. Getting a t5-small model (60m parameters) to outperform phi-3.5-mini (3.8B) and mistral (7B) based on data generated without any supervision is hardly marginal.
>
> | System   | Python |           | Excel  |           | Powershell |           |
> |----------|--------|-----------|--------|-----------|------------|-----------|
> |          | Sketch | Execution | Sketch | Execution | Sketch     | Execution |
> | RING     | 0.78   | 0.82      | 0.68   | 0.74      | 0.15       | 0.18      |
> | GPT-4o   | 0.81   | 0.84      | 0.68   | 0.75      | 0.15       | 0.24      |
> | LaMirage | 0.67   | 0.71      | 0.69   | 0.72      | -          | -         |
> | BiFi     | 0.72   | 0.76      | -      | -         | -          | -         |
> | Codex    | 0.56   | 0.6       | 0.65   | 0.67      | 0.08       | 0.1       |
>
> \
> **Question-2: Incomplete Explanation on GPT-4o Prompting**
>
> We will add our prompt and a sample of data to the supplementary material. Note that we only use the model to generate synthetic training data (to break a formula)—not to repair—so it might not benefit from reasoning. Our prompt is a few-shot prompt where we instruct the model to break a formula a human might generate. We do 2 versions of this - (1) We use the syntax operators listed in BiFi [1] and (2) We do not provide guidance so it can cover all types of errors.
>
> \
> **Question-3: *Unexplored Trade-Offs:* The paper removes NL component from the CodeFusion model, turning it into an unconditional model.**
>
> CodeFusion did not use NL in the pre-training step, and we are only leveraging the pre-training step. It’s just an artefact of what we are exploring: whether an unconditional code diffusion model, which yields noisy code in intermediate steps that resemble buggy code written by humans, can be leveraged to repair that buggy code or not.
> Repairing code with a natural language instruction (generated by a model or as a hint provided by the user) is not (supposed to be) part of our current exploration.
> The main contribution of our work is to show that diffusion models have such latent repair capabilities (without supervising for repair)
>
> \
> **Question-4: What are the Y-axes representing in Figure 6?**
>
> Figure 6 has distribution plots and the y-axis has the frequency of the bins. We should have specified that these are distribution plots and will update the description.
>
> \
> **Question-5: The authors claim that the diffusion process mirrors human debugging behavior. However, there are no concrete qualitative examples supporting this**
>
> We will clarify this claim. We show if buggy code is close enough to the types of bugs encountered during reverse diffusion, the model can use its own strategy to arrive at the correct code. We will both update this claim and add some more examples of the diffusion process to the paper. We will also add more citations on how humas repair code and connecting these patterns to diffusion [1].
>
> [1]: Where to Look When Repairing Code?, Dominik Huber and Matteo Paltenghi and Michael Pradel.
>
> \
> **Question-6: Are the number of tokens taken by the decoder same as the final generation?**
>
> In actual implementation, the decoder is trained to generate N tokens as well, one of which is an end-of-sequence tokens and the rest are padding tokens. We will clarify these details in Section 2.
>
> \
> **Question-7: Differentiate from CodeFusion (Singh et al., 2023)**
>
> Our work is tangential to CodeFusion as we explore the latent repair capabilities of diffusion models which has not been studied before.
> The main contribution of our work is to show that diffusion models have such latent repair capabilities (without supervising for repair) which is not explored and is very interesting for these models.
>
> \
> **Question-8: Open Source**
>
> Yes. Both the model and generated data will be open-sourced.

---

> ### Comment · Reviewer_2pXc · 2024-11-26
>
> > Getting a t5-small model (60m parameters) to outperform phi-3.5-mini (3.8B) and mistral (7B) based on data generated without any supervision is hardly marginal.
>
> I appreciate your clarification that your approach focuses on the latent repair capabilities of the diffusion model. However, the observation that a small model like t5-small outperforms a model more than 100 times larger, such as Mistral 7B, when fine-tuned on the same synthetic data, is highly counterintuitive. A model of Mistral 7B’s scale should, in principle, leverage the same data far more effectively due to its capacity and generalization abilities. This discrepancy raises concerns about either the quality of the generated data or the experimental setup. Could you provide further insights into why this might be happening?
>
> > Note that we only use the model to generate synthetic training data (to break a formula)—not to repair—so it might not benefit from reasoning.
>
> While I understand your point that GPT-4o was used to generate synthetic bugs and not for repair tasks, I still find the restriction to basic syntactic errors problematic. Generating realistic bugs intuitively requires reasoning to mimic the diverse and nuanced mistakes humans make, and larger models like GPT-4o are naturally better suited for this due to their extensive training. Limiting GPT-4o to syntactic errors feels like an artificial restriction rather than a genuine limitation of the model. This choice also seems to undercut the potential of GPT-4o to generate more diverse and complex errors, which would make for richer training data. Including more details about the prompting setup, as you plan to do, is essential to clarify this and ensure reproducibility.
>
> > We will clarify this claim. We show if buggy code is close enough to the types of bugs encountered during reverse diffusion, the model can use its own strategy to arrive at the correct code.
>
> Your revised explanation—that the model succeeds when buggy code aligns with reverse diffusion patterns—is significantly different from the original claim in the paper that the diffusion process mirrors human debugging behavior. If this adjustment represents a core shift in the narrative, the paper’s title and framing should also reflect this change to avoid misrepresenting the scope or contributions. A clearer alignment between claims, results, and framing will make the paper easier to follow and review.
>
> > Yes. Both the model and generated data will be open-sourced.
>
> I am glad to hear that the model and generated data will be open-sourced. This step will aid reproducibility and future research.

---

> > ### Author Response · Authors · 2024-11-28
> > **Rebuttal**
> >
> > **Question-1:**
> > ```
> > I appreciate your clarification that your approach focuses on the latent repair capabilities of the diffusion model. However, the observation that a small model like t5-small outperforms a model more than 100 times larger, such as Mistral 7B, when fine-tuned on the same synthetic data, is highly counterintuitive. A model of Mistral 7B’s scale should, in principle, leverage the same data far more effectively due to its capacity and generalization abilities. This discrepancy raises concerns about either the quality of the generated data or the experimental setup. Could you provide further insights into why this might be happening?`
> > ```
> >
> > Thank you for bringing attention to this mistake at our end. We had confused the back-end model for CodeFusion (codet5-small) and the model fine-tuned to repair, which is a CodeT5+ 2B model. Since this is a reasonably sized model that was trained on code on tasks like masked language modeling, its strong performance for repair is perfectly understandable. We have updated all references in the paper and mentioned full model versions for all models.
> > Note that we only use the model to generate synthetic training data (to break a formula)—not to repair—so it might not benefit from reasoning.
> >
> > \
> > \
> > **Question-2:**
> > ```
> > While I understand your point that GPT-4o was used to generate synthetic bugs and not for repair tasks, I still find the restriction to basic syntactic errors problematic. Generating realistic bugs intuitively requires reasoning to mimic the diverse and nuanced mistakes humans make, and larger models like GPT-4o are naturally better suited for this due to their extensive training.
> > ```
> >
> > Limiting GPT-4o to syntactic errors feels like an artificial restriction rather than a genuine limitation of the model. This choice also seems to undercut the potential of GPT-4o to generate more diverse and complex errors, which would make for richer training data. Including more details about the prompting setup, as you plan to do, is essential to clarify this and ensure reproducibility.
> > The scope of errors was based on both related work (BIFI, FLAME) and the scope of errors that the benchmarks were designed for (last-mile repairs). We have extended the description of how training data was generated and added the prompt to the Appendix. Note that we did allow the model to use chain-of-thought while generating data (as you can see in the prompt). In our earlier response, we meant that we did not generate “data with reasoning traces” to then fine-tune a model in a chain-of-thought way.
> >
> > In practice, one would combine the data generated by diffusion (with focus on diversity and syntax) and by GPT-4o (for complex errors that require more intuition about user errors). The experiments in our paper are designed to show that the data generated by diffusion does hold significant potential. We have added a Venn-diagram that shows the overlap between solved cases for models fine-tuned on data from different sources to highlight this potential for combining data from different sources.
> > We will clarify this claim. We show if buggy code is close enough to the types of bugs encountered during reverse diffusion, the model can use its own strategy to arrive at the correct code.
> >
> > \
> > \
> > **Question-3:**
> > ```
> > Your revised explanation—that the model succeeds when buggy code aligns with reverse diffusion patterns—is significantly different from the original claim in the paper that the diffusion process mirrors human debugging behavior. If this adjustment represents a core shift in the narrative, the paper’s title and framing should also reflect this change to avoid misrepresenting the scope or contributions. A clearer alignment between claims, results, and framing will make the paper easier to follow and review.
> > ```
> >
> > Based on feedback from reviewers, we have verified that claims made in the paper are aligned with the experiments that we performed: the potential of applying diffusion to last-mile repair. We believe that the title still holds: if the model can generate code that overlaps enough with human errors, regardless of the behavior of how it got there, it can still be considered an operator that applies human noise to code.

---

> ### Author Response · Authors · 2024-12-02
> **Rebuttal**
>
> We hope we were able to answer the concerns raised. Since today is the last day for submitting questions regarding our paper, we wanted to check in if there are any further clarifications. We appreciate your time, and the valuable feedback provided, which was valuables to improving our work.
>
> We have updated the manuscript to address the concerns raised by reviewers. As a brief summary, we have significantly enhanced the narrative of our paper, emphasizing the applications of diffusion for last-mile repair. We have made a range of improvements, including revisiting our claims, adding results for comparison, and including new visualizations to support our findings. The general comment describes the changes in greater detail
>
> If you have any further questions or require clarification on any aspect of the paper, we would be more than happy to address them before the deadline.

---

### Official Review · Reviewer_awoK · 2024-11-04

**Soundness:** 2
**Presentation:** 3
**Contribution:** 2
**Rating:** 5
**Confidence:** 3

**Summary:**

The authors explore the potential of diffusion models in code generation, focusing on the iterative process of removing noise from the latent representation of a code snippet. As the diffusion process approaches convergence, the resulting edits resemble "last-mile repairs" applied to broken or incomplete code. This study evaluates the similarities between errors encountered by these models and those faced by human developers, assessing the models' capabilities in performing last-mile repairs. The authors propose two impactful applications: first, leveraging the diffusion model for last-mile repair by adding noise to a broken code snippet and continuing the diffusion process; second, utilizing the model to generate an arbitrary amount of training data for other last-mile repair methods, which are computationally more efficient, by sampling intermediate and final program states from the diffusion pipeline. Experiments are conducted to evaluate both applications, alongside an analysis of trends in representation evolution throughout the diffusion process, offering valuable insights into the underlying reasoning.

**Strengths:**

+ Novel idea.

**Weaknesses:**

- Focus on Program Repair.
- Speed of Diffusion Models.
- Lack of Baselines.
- Closed Source.

**Questions:**

The authors present an innovative approach by combining diffusion models with program repair, highlighting the unique potential of this technique in enhancing code generation and error correction.

I have four concerns.

1. Focus on Program Repair.

The authors claim that their exploration has two main implications: using the diffusion model for direct code repair and generating training data for specialized approaches. However, the latter application is evaluated exclusively in the context of program repair. This narrow focus raises questions about the broader applicability of their findings, especially given that the paper is titled "CODE DIFFUSION MODELS ARE CONTINUOUS HUMAN NOISE OPERATORS." It would benefit the authors to emphasize the program repair aspect more clearly and consider the implications of their work beyond this specific domain.

2. Speed of Diffusion Models.

The authors should address the fact that diffusion models tend to be slower than other program repair approaches. A comparative analysis of the speed and efficiency of their method versus existing techniques (e.g., the most similar model without diffusion or just remove the diffusion of your approach) would provide valuable context for evaluating its practicality.

3. Lack of Baselines.

The paper does not adequately consider baseline models to demonstrate the effectiveness of the diffusion approach. Including comparisons with established methods (e.g., the baselines used in "https://dl.acm.org/doi/pdf/10.1145/3650212.3680323") or the baselines of the original model in Table1 would strengthen the argument for the proposed model's advantages and validate its performance claims.

4. Closed Source.

 The closed-source nature of the implementation may limit the reproducibility of results and the ability for the community to build upon this work.

---

> ### Author Response · Authors · 2024-11-22
> **Rebuttal**
>
> **Question-1: *Focus on Program Repair:* It would benefit the authors to emphasize the program repair aspect more clearly and consider the implications of their work beyond this specific domain of repair.**
>
> We meant generating training data for specialized repair approaches and will clarify that in the text. Our claims are specifically about code repair, where the inverse of repairing code is breaking code—introducing noise. This noise is typically introduced in the discrete token space, by implementing specialized functions that imitate human errors [1, 2]. Our aim is to show that the diffusion process—iteratively removing noise from a latent (continuous) representation—shares enough similarities to the discrete errors that humans make to be useful for repair.
> - [1] https://arxiv.org/abs/2106.06600
> - [2] https://proceedings.mlr.press/v119/yasunaga20a.html
>
> \
> **Question-2: *Speed of Diffusion Models:* The authors should address diffusion models tend to be slower than other program repair approaches. A comparative analysis of the speed of the method versus existing techniques (e.g., the most similar model without diffusion or just remove the diffusion of your approach) would provide valuable context for evaluating its practicality.**
>
> Diffusion models are indeed slower, due to their iterative nature, which we will explicitly mention in the limitation section, along with some potential solutions. This concern motivated us to use the diffusion process to generate data for specialized repair approaches, where codet5-small yields excellent results (Table 2). Note that we are not claiming that, in its current state, diffusion is state-of-the-art for code repair: we are investigating a foundational insight into the potential applications of diffusion for repair. Exploiting these insights into a practical solution leaves a significant potential for future work.
>
> We have included the latency and memory results for Diffusion and Autoregressive techniques and are happy to add this to the paper. To reiterate, we are not claiming that diffusion is state-of-the-art for code repair, rather we are exploring the insight that diffusion models have latent repair abilities which is very interesting.
>
> | Approach      | Latency | Memory |
> |---------------|---------|--------|
> | Diffusion-60m | 2931    | 424    |
> | Phi-3.8b      | 543     | 764    |
> | Phi-7b        | 771     | 891    |
> | Phi-14b       | 1093    | 994    |
>
> \
> **Question-3: *Lack of Baselines:* The paper does not consider baseline models to demonstrate the benefits of the diffusion approach (e.g., the baselines used in "https://dl.acm.org/doi/pdf/10.1145/3650212.3680323").**
>
> We do not claim that, in its current state, diffusion should be used as a practical repair approach that achieves state-of-the-art results. Our paper makes fundamental observations about repair behavior of a reverse diffusion process for code, showing that (1) it can repair syntax errors and (2) it can be used to generate training data for a specialized model.
>
> To make these insights stronger and provide insights into the complexity of the benchmarks, we can report the performance of existing repair approaches on these benchmarks. For example, the RING [1] paper that introduced the PowerShell benchmarks, reports a performance of 10% in pass@1 and 27% in pass@50 using the 175B parameter Codex model. We show a repair performance of 18% using pooled diffusion and 34.2% on fine-tuning a codet5-small (60M parameters) on synthetic data.
>
> We have included the results for different baselines for repair as reference to how they compare to Diffusion and are happy to add this to the paper. We would again like to highlight the aim of this paper is not to say that diffusion models in its current state are the SOTA in code repair, rather, the paper investigates the latent repair abilities of diffusion models (without training them for this task) and especially since these have not been considered in the context of repair.
>
> | System   | Python |           | Excel  |           | Powershell |           |
> |----------|--------|-----------|--------|-----------|------------|-----------|
> |          | Sketch | Execution | Sketch | Execution | Sketch     | Execution |
> | RING     | 0.78   | 0.82      | 0.68   | 0.74      | 0.15       | 0.18      |
> | GPT-4o   | 0.81   | 0.84      | 0.68   | 0.75      | 0.15       | 0.24      |
> | LaMirage | 0.67   | 0.71      | 0.69   | 0.72      | -          | -         |
> | BiFi     | 0.72   | 0.76      | -      | -         | -          | -         |
> | Codex    | 0.56   | 0.6       | 0.65   | 0.67      | 0.08       | 0.1       |
>
> \
> **Question-4: *Closed Source:* The closed-source nature of the implementation may limit the reproducibility of results and the ability for the community to build upon this work.**
>
> We plan to fully open source our code, models and the generated training data. We have not added the GitHub url for anonymity but will add it in the final version.

---

> > ### Comment · Reviewer_awoK · 2024-11-27
> > **Still 5.**
> >
> > The question 1&4 are minors. However, for the question 2, the comparison between Diffusion-60m with the larger models Phi-xxb is unfair. I prefer to keep my original score.

---

### Author Response · Authors · 2024-11-28
**General Rebuttal**

We would like to thank the reviewers for their valuable feedback, which we have used to significantly improve the presentation and narrative of our paper, which now emphasizes the fact diffusion has promising applications to last-mile repair.

In the updated PDF, these changes are marked in red. We provide a brief overview of the most important improvements.

- We have **revisited claims about diffusion mirroring human behavior**, emphasizing that the similarity between errors encountered during the reverse diffusion process and those made by humans enables the diffusion model to (1) directly perform last-mile repair and (2) generate training data for fine-tuning last-mile repair models.
- We have moved the experiment section on trends in code refinement to Section 2, to motivate our application of diffusion to last-mile repair.
- We have improved the paragraph that describes the CodeFusion model and ablated versions used in our evaluation.
- To put the potential of diffusion for repair into perspective, we have **added results for different baselines on our benchmarks**. Whereas the diffusion model does not perform better than specialized approaches and GPT-4o, it does perform better than the Codex model on Python and PowerShell, which is a remarkable result.
- To emphasize the diversity benefit of diffusion, **we added pass@1, pass@3 and pass@5 rates**, where the improvements for diffusion (+4% -- 5%) are much more significant than those for GPT-4o and RING (+ < 1%).
- We added a more **complete description and motivation of baseline** approaches for generating synthetic data and added the template of our prompt to the Appendix.
- We fixed the reported models for fine-tuning, where CodeT5+ (2B) was used instead of CodeT5-small (60M), and would like to apologize for any confusion caused.
- We added a Venn-diagram that compares the overlap in Excel benchmarks solved using models fine-tuned on different sources, to highlight the potential of combining data from GPT-4o and diffusion. Note that **diffusion completely supersedes the syntactic data operators.**
- We have explicitly added a limitation about the speed of diffusion models for practical applications. We have also highlighted that we are not proposing diffusion to replace repair systems rather we explore its abilities for latent repair which have not been explored. **Based on the current results, we have also made it explicit that using diffusion to generate synthetic data shows the most promise, so the latency of diffusion does not diminish our contributions.**

---

### Meta-Review · Area_Chair_fR2L · 2024-12-21

**Metareview:**

This paper presents an approach leveraging diffusion models to understand noise in source code. Towards the end of backward inference in a diffusion model, the state of a code sequence can be interpreted as a "buggy" version of that source code: are these bugs similar enough to human bugs to make this a useful analogy? This enables two applications: diffusion models for code repair and generation of training data for other approaches (by treating the diffusion model as a density function over buggy code).

This paper presents an interesting and novel application of diffusion models for code. The experiments cover multiple programming languages. After some edits, the paper is generally well-written.

The results on code repair are mixed. Other approaches like LLMs (which have many more parameters) are able to do substantially better on 2 of the 3 tasks. However, this approach works well on PowerShell, and as the authors point out, comparing to LLMs isn't really a fair direct comparison. Using the diffusion model as a noise process works better, though the baselines compared to here are fairly rudimentary.

There is also a question of whether the main claim of the title is supported: P1bD, 2pXc, and VztF raise the question about whether it is really shown that *human* errors and debugging behavior are mimicked. I think this is important to show, especially because the authors claim that they are not pushing novel methodology or SOTA performance. In my view, there should be more analysis of this claim beyond the somewhat black-box training approaches taken here.

The approach is slower than other models of similar size (rebuttal to awoK).

**Additional Comments On Reviewer Discussion:**

There were several important points of the response. The speed results in response to awoK are helpful to see, as are the additional baselines.

I think the point raised by 2pXc is also very relevant.

> Your revised explanation—that the model succeeds when buggy code aligns with reverse diffusion patterns—is significantly different from the original claim in the paper that the diffusion process mirrors human debugging behavior. If this adjustment represents a core shift in the narrative, the paper’s title and framing should also reflect this change to avoid misrepresenting the scope or contributions. A clearer alignment between claims, results, and framing will make the paper easier to follow and review."

Finally, the authors claim

> We do not claim to present a new method. We show that a diffusion model can be used out-of-the box to generate training data for code syntax repair, because the distribution of “broken” code encountered during training has some overlap with the distribution of syntax mistakes that humans make. Thus, we do not train the diffusion model on repair and only show latent repair capabilities of the models.

These clarify the intent of the paper quite a bit. However, I think the interest of this paper still depends on how surprising this method is and how well it works. In my view, the method should either making deeper claims about mimicking the human distribution *or* show slightly stronger experimental results.

---

### Decision · Program_Chairs · 2025-01-22

Reject